# Toward Robust Incomplete Multimodal Sentiment Analysis via Hierarchical Representation Learning

**Mingcheng Li**[1,3*]   **Dingkang Yang**[1,3*†]   **Yang Liu**[1]   **Shunli Wang**[1,3]   **Jiawei Chen**[1,3]
**Shuaibing Wang**[1,3]   **Jinjie Wei**[1,3]   **Yue Jiang**[1,3]   **Qingyao Xu**[1,3]   **Xiaolu Hou**[1,3]
**Mingyang Sun**[1,3]   **Ziyun Qian**[1,3]   **Dongliang Kou**[1,3]   **Lihua Zhang**[1,2,3,4,5†]

[1] Academy for Engineering and Technology, Fudan University, Shanghai, China
[2] Institute of Metaverse & Intelligent Medicine, Fudan University, Shanghai, China
[3] Cognition and Intelligent Technology Laboratory, Shanghai, China
[4] Jilin Provincial Key Laboratory of Intelligence Science and Engineering, Changchun, China
[5] Engineering Research Center of AI and Robotics, Ministry of Education, Shanghai, China.
`mingchengli21@m.fudan.edu.cn, dkyang20@fudan.edu.cn`

## Abstract

Multimodal Sentiment Analysis (MSA) is an important research area that aims to understand and recognize human sentiment through multiple modalities. The complementary information provided by multimodal fusion promotes better sentiment analysis compared to utilizing only a single modality. Nevertheless, in real-world applications, many unavoidable factors may lead to situations of uncertain modality missing, thus hindering the effectiveness of multimodal modeling and degrading the model's performance. To this end, we propose a Hierarchical Representation Learning Framework (HRLF) for the MSA task under uncertain missing modalities. Specifically, we propose a fine-grained representation factorization module that sufficiently extracts valuable sentiment information by factorizing modality into sentiment-relevant and modality-specific representations through crossmodal translation and sentiment semantic reconstruction. Moreover, a hierarchical mutual information maximization mechanism is introduced to incrementally maximize the mutual information between multi-scale representations to align and reconstruct the high-level semantics in the representations. Ultimately, we propose a hierarchical adversarial learning mechanism that further aligns and adapts the latent distribution of sentiment-relevant representations to produce robust joint multimodal representations. Comprehensive experiments on three datasets demonstrate that HRLF significantly improves MSA performance under uncertain modality missing cases.

## 1   Introduction

Multimodal sentiment analysis (MSA) has attracted wide attention in recent years. Unlike unimodal emotion recognition tasks [9, 63, 64, 53, 56], MSA understands and recognizes human emotions through multiple modalities, including language, audio, and visual [31, 58]. Previous studies have shown that combining complementary information among different modalities facilitates valuable semantic generation [41, 40, 61, 55, 62]. MSA has been well studied so far under the assumption that all modalities are available in the training and inference phases [12, 66, 54, 57, 56, 25, 59, 60]. Nevertheless, in real-world applications, modalities may be missing due to security concerns, background noises, sensor limitations and so on. Ultimately, these incomplete multimodal data significantly hinder the performance of MSA. For instance, as shown in Figure 1, the entire visual

---

*Equal contributions. †Corresponding authors.

38th Conference on Neural Information Processing Systems (NeurIPS 2024).

modality and some frame-level features in the language and audio modalities are missing, leading to an incorrect prediction.

In recent years, many studies [8, 28, 26, 49, 37, 50, 76, 74, 68, 27, 23, 22] attempt to address the problem of missing modalities in MSA. For example, SMIL [29] estimates the latent features of the missing modality data via Bayesian Meta-Learning. However, these methods are constrained by the following factors: **(i)** Implementing complex feature interactions for incomplete modalities leads to a large amount of information redundancy and cumulative errors, resulting in ineffective extraction of sentiment semantics. **(ii)** Lacking consideration of semantic and distributional alignment of representations, causing imprecise feature reconstruction and nonrobust joint representations.

To address the above issues, we propose a Hierarchical Representation Learning Framework (HRLF) for the MSA task under uncertain missing modalities. HRLF has three core contributions: **(i)** We present a fine-grained representation factorization module that sufficiently extracts valuable sentiment information by factorizing modality into sentiment-relevant and modality-specific representations through intra- and inter-modality translations and sentiment semantic reconstruction. **(ii)** Furthermore, a hierarchical mutual information maximization mechanism is introduced to incrementally align the high-level semantics by maximizing the mutual information of the multi-scale representations of both networks in knowledge distillation. **(iii)** Eventually, we propose a hierarchical adversarial learning mechanism to progressively align

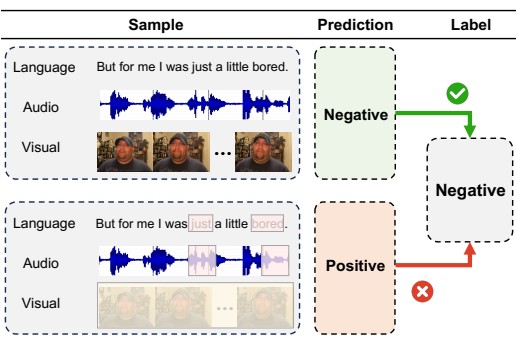

Figure 1: A case of incorrect prediction by the traditional model with missing modalities. The pink and yellow areas indicate intra- and inter-modality missingness, respectively.

the latent distributions of representations leveraging multi-scale adversarial learning. Based on these components, HRLF significantly improves MSA performance under uncertain modality missing cases on three multimodal benchmarks.

## 2 Related Work

### 2.1 Multimodal Sentiment Analysis

Multimodal Sentiment Analysis (MSA) seeks to comprehend and analyze human sentiment by utilizing diverse modalities. Unlike conventional single-modality sentiment recognition, MSA poses greater challenges owing to the intricate nature of processing and analyzing heterogeneous data across modalities. Mainstream studies in MSA [69, 70, 44, 12, 11, 42, 25] focus on designing complex fusion paradigms and interaction mechanisms to improve MSA performance. For instance, CubeMLP [42] employs three distinct multi-layer perceptron units for feature amalgamation along three axes. However, these methods rely on complete modalities and thus are impractical for real-world deployment. There are two primary approaches for addressing the missing modality problem in MSA: (1) Generative methods [8, 28, 26, 49] and (2) joint learning methods [37, 50, 76, 74, 68, 27]. Generative methods aim to regenerate missing features and semantics within modalities by leveraging the distributions of available modalities. For example, TFR-Net [67] employs a feature reconstruction module to guide the extractor to reconstruct missing semantics. Joint learning methods focus on deriving cohesive joint multimodal representations based on inter-modality correlations. For instance, MMIN [76] produces robust joint multimodal representations via cross-modality imagination. However, these methods cannot extract rich sentiment information from incomplete modalities due to their inefficient interaction. In contrast, our learning paradigm achieves effective extraction and precise reconstruction of sentiment semantics through complete modality factorization.

### 2.2 Factorized Representation Learning

The fundamental goal of learning factorized representations is to disentangle representations that have different semantics and distributions. This separation enables the model to more effectively capture intrinsic information and yield favorable modality representations. Previous methods of factorized

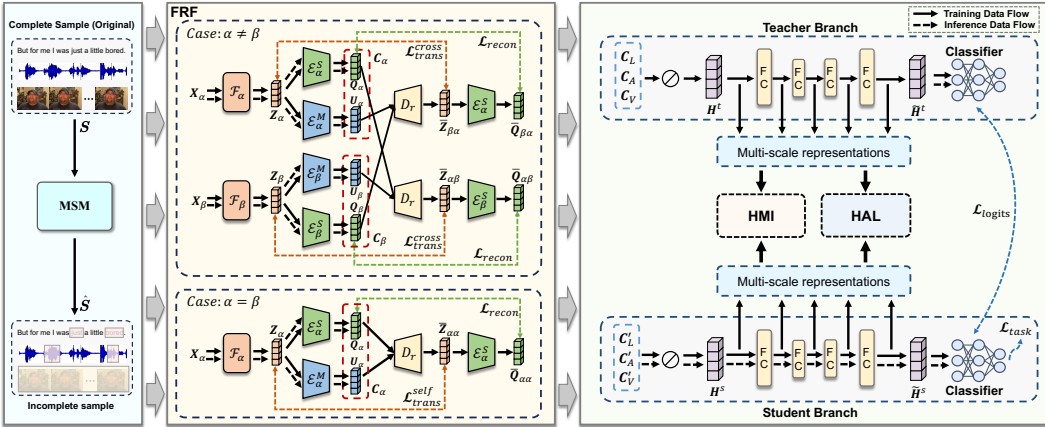

Figure 2: The structure of our HRLF, which consists of three core components: Fine-grained Representation Factorization (FRF) module, Hierarchical Mutual Information (HMI) maximization mechanism, and Hierarchical Adversarial Learning (HAL) mechanism.

representation learning primarily rely on auto-encoders [3] and generative adversarial networks [32]. For example, FactorVA [18] is introduced to achieve factorization by leveraging the characteristic that representations are both factorial and independent in dimension. Recently, factorization learning has been progressively utilized in MSA tasks [54, 25, 57]. For instance, FDMER [54] utilizes consistency and discreteness constraints between modalities to disentangle modalities into modality-invariant and modality-private features. DMD [25] disentangles each modality into modality-independent and modality-exclusive representations and then implements a knowledge distillation strategy among the representations with dynamic graphs. MFSA [57] refines multimodal representations and learns complementary representations across modalities by learning modality-specific and modality-agnostic representations. Despite the progress these studies have brought to MSA, certain limitations persist: (i) The supervision of the factorization process is coarse-grained and insufficient. (ii) Focusing solely on factorizing distinct representations at the modality level, without taking into account sentimentally beneficial and relevant representations. By contrast, the proposed method decomposes sentiment-relevant representations precisely through intra- and inter-modality translation and sentiment semantic reconstruction. Furthermore, hierarchical mutual information maximization and adversarial learning paradigms are employed to refine and optimize the representation of factorization at the semantic level and the distributional level, respectively, thus yielding robust joint multimodal representations.

## 2.3 Knowledge Distillation

Knowledge distillation leverages additional supervisory signals from a pre-trained teacher network to aid in training a student network [15]. There are generally two categories of knowledge distillation methods: distillation from intermediate features [13, 14, 19, 33, 35, 39, 45, 43, 65, 73] and distillation from logits [6, 10, 30, 52, 75]. Many studies [5, 17, 38, 21, 47, 51] utilize knowledge distillation for MSA tasks with missing modalities. These approaches aim to transfer dark knowledge from teacher networks trained on complete modalities to student networks trained by missing modalities. The teacher network typically provides richer and more comprehensive feature representations than the student network. For instance, KD-Net [17] utilizes a teacher network with complete modalities to supervise the unimodal student network at both the feature and logits levels. Despite their promising results, these methods neglect precise supervision of representations, resulting in low-quality knowledge transfer. To this end, we implement hierarchical semantic and distributional alignment of the multi-scale representations of both networks to transfer knowledge effectively.

# 3 Methodology

## 3.1 Problem Formulation

Given a multimodal video segment with three modalities as $S = [X_L, X_A, X_V]$, where $X_L \in \mathbb{R}^{T_L \times d_L}$, $X_A \in \mathbb{R}^{T_A \times d_A}$, and $X_V \in \mathbb{R}^{T_V \times d_V}$ denote language, audio, and visual modalities,

respectively. $\mu = \{L, A, V\}$ denotes the set of modality types. $T_m(\cdot)$ is the sequence length and $d_m(\cdot)$ is the embedding dimension, where $m \in \mu$. We define two missing modality cases to simulate the most natural and holistic challenges in real-world scenarios: (1) *intra-modality missingness*, which indicates some frame-level features in the modality sequences are missing. (2) *inter-modality missingness*, which denotes some modalities are entirely missing. We aim to recognize the utterance-level sentiments using incomplete multimodal data.

## 3.2 Overall Framework

Figure 2 illustrates the main workflow of HRLF. The teacher and student networks adopt a consistent structure but have different parameters. During the training phase, the workflow of our HRLF is as follows: (i) We first train the teacher network with complete-modality samples and their sentiment labels. (ii) Given a video segment sample $\boldsymbol{S}$, we generate a missing-modality sample $\hat{\boldsymbol{S}}$ with the Modality Stochastic Missing (MSM) strategy. MSM simultaneously performs intra-modality missingness and inter-modality missingness. $\boldsymbol{S}$ and $\hat{\boldsymbol{S}}$ are fed into the pre-trained teacher network and the initialized student network, respectively. (iii) We input each sample into the FRF module, to factorize each modality into a sentiment-relevant representation $\boldsymbol{Q}_m$ and a modality-specific representation $\boldsymbol{U}_m$, where $m \in \mu$. (iv) Sequences $[\boldsymbol{C}_L, \boldsymbol{C}_A, \boldsymbol{C}_V]$ and $[\boldsymbol{C}'_L, \boldsymbol{C}'_A, \boldsymbol{C}'_V]$ are generated by concatenating $\boldsymbol{Q}_m$ and $\boldsymbol{U}_m$ from all modalities in the teacher and student networks. Each element of the sequences is concatenated to yield the joint multimodal representations $\boldsymbol{H}^t$ and $\boldsymbol{H}^s$. (v) The multi-scale representations of both networks are obtained by passing $\boldsymbol{H}^t$ and $\boldsymbol{H}^s$ through the fully-connected layers. The proposed HMI and HAL are used to align the semantics and distribution between the multiscale representations. (vi) The outputs $\tilde{\boldsymbol{H}}^t$ and $\tilde{\boldsymbol{H}}^s$ of the fully-connected layers are fed into the task-specific classifier to get logits $\boldsymbol{L}^t$ and $\boldsymbol{L}^s$. We constrain the consistency between logits and utilize $\boldsymbol{L}^s$ to implement the sentiment prediction. In the inference phase, testing samples are only fed into the student network for downstream tasks.

## 3.3 Fine-grained Representation Factorization

Modality missing leads to ambiguous sentiment cues in the modality and information redundancy in multimodal fusion. It hinders the model from capturing valuable sentiment semantics and filtering sentiment irrelevant information. Although previous studies in MSA [12, 54] decompose the task-relevant semantics contained in the modality to some extent via simple auto-encoder networks with reconstruction constraints, their purification of sentiment semantics is inadequate, and they cannot be applied to modality missing scenarios. Therefore, we propose a Fine-grained Representation Factorization (FRF) module to capture sentiment semantics in modalities. The core idea is to factorize each modality representation into two types of representations: (1) sentiment-relevant representation, which contains the holistic sentiment semantics of the sample. It is modality-independent, shared across all modalities of the same subject, and robust to modality missing situations. (2) modality-specific representation, which represents modality-specific task-independent information.

As shown in Figure 2, FRF receives the multimodal sequences $[\boldsymbol{X}_L, \boldsymbol{X}_A, \boldsymbol{X}_V]$ with modality number $n = 3$. The modality $\boldsymbol{X}_\alpha$ with $\alpha \in \mu$ passes through a 1D temporal convolutional layer with kernel size $3 \times 3$ and adds the positional embedding [46] to obtain the preliminary representations, denoted as $\boldsymbol{R}_\alpha = \boldsymbol{W}_{3 \times 3}(\boldsymbol{X}_\alpha) + PE(T_\alpha, d) \in \mathbb{R}^{T_\alpha \times d}$. The $\boldsymbol{R}_\alpha$ is fed into a Transformer [46] encoder $\mathcal{F}_\alpha(\cdot)$, and the last element of its output is denoted as $\boldsymbol{Z}_\alpha = \mathcal{F}_\alpha(\boldsymbol{R}_\alpha) \in \mathbb{R}^d$. The $\boldsymbol{Z}_\alpha \in \mathcal{Z}_\alpha$ is the low-level modality representation of the modality $\alpha$. We aim to factorize modality representation $\boldsymbol{Z}_\alpha$ into a sentiment-relevant representation $\boldsymbol{Q}_\alpha$ by a sentiment encoder $\boldsymbol{Q}_\alpha = \mathcal{E}_\alpha^S(\boldsymbol{Z}_\alpha)$ and a modality-specific representation $\boldsymbol{U}_\alpha$ by a modality encoder $\boldsymbol{U}_\alpha = \mathcal{E}_\alpha^M(\boldsymbol{Z}_\alpha)$. $\mathcal{E}_\alpha^S(\cdot)$ and $\mathcal{E}_\alpha^M(\cdot)$ are composed of multi-layer perceptrons with the ReLU activation. The following two processes ensure adequate factorization and semantic reinforcement of the above two representations.

**Intra- and Inter-modality Translation.** The proposed FRF effectively decouples sentiment-relevant and modality-specific representations by simultaneously performing intra- and inter-modality translations. Given a pair of representations $\boldsymbol{Q}_\alpha$ and $\boldsymbol{U}_\beta$ factorized by $\boldsymbol{Z}_\alpha$ and $\boldsymbol{Z}_\beta$ with $\alpha, \beta \in \mu$, the decoder $\mathcal{D}_r(\cdot)$ is supposed to translate and synthesize the representation $\overline{\boldsymbol{Z}}_{\alpha\beta}$, whose reconstructed domain corresponds to the modality representation $\boldsymbol{Z}_\beta \in \mathcal{Z}_\beta$. The $\mathcal{D}_r(\cdot)$ consists of feed-forward neural layers. The modality translations include intra-modality translation (*i.e.*, $\alpha = \beta$) and inter-modality translation (*i.e.*, $\alpha \neq \beta$), whose losses are respectively denoted as:

$$\mathcal{L}_{trans}^{self} = \frac{1}{n} \sum_{\alpha \in \mu} \mathbf{E}_{\boldsymbol{Z}_\alpha \sim \boldsymbol{\mathcal{Z}}_\alpha} \left[ \left\| \overline{\boldsymbol{Z}}_{\alpha\alpha} - \boldsymbol{Z}_\alpha \right\|_2 \right], \tag{1}$$

$$\mathcal{L}_{trans}^{cross} = \frac{1}{n^2 - n} \sum_{\alpha \in \mu} \sum_{\beta \in \mu, \beta \neq \alpha} \mathbf{E}_{\boldsymbol{Z}_\alpha \sim \boldsymbol{\mathcal{Z}}_\alpha, \boldsymbol{Z}_\beta \sim \boldsymbol{\mathcal{Z}}_\beta} \left[ \left\| \overline{\boldsymbol{Z}}_{\alpha\beta} - \boldsymbol{Z}_\beta \right\|_2 \right], \tag{2}$$

where $\overline{\boldsymbol{Z}}_{\alpha\beta} = \mathcal{D}_r(\mathcal{E}_\alpha^S(\boldsymbol{Z}_\alpha), \mathcal{E}_\beta^M(\boldsymbol{Z}_\beta))$. The translation loss is denoted as: $\mathcal{L}_{trans} = \mathcal{L}_{trans}^{self} + \mathcal{L}_{trans}^{cross}$.

**Sentiment Semantic Reconstruction.** To ensure that the reconstructed modality still contains the sentiment semantics from the original modality, we encourage both to maintain the consistency of sentiment-relevant semantics and utilize the following loss for constraints, denoted as:

$$\mathcal{L}_{recon} = \frac{1}{n^2} \sum_{\alpha \in \mu} \sum_{\beta \in \mu} \mathbf{E}_{\boldsymbol{Z}_\alpha \sim \boldsymbol{\mathcal{Z}}_\alpha, \boldsymbol{z}_\beta \sim \boldsymbol{\mathcal{z}}_\beta} \left[ \left\| \overline{\boldsymbol{Q}}_{\beta\alpha} - \boldsymbol{Q}_\alpha \right\|_2 \right], \tag{3}$$

where $\overline{\boldsymbol{Q}}_{\beta\alpha} = \mathcal{E}_\alpha^S \left( \mathcal{D}_r \left( \mathcal{E}_\beta^S (\boldsymbol{Z}_\beta), \mathcal{E}_\alpha^M (\boldsymbol{Z}_\alpha) \right) \right)$ is the sentiment-relevant representation derived from the reconstructed modality representation. Consequently, the final loss of the FRF is denoted as:

$$\mathcal{L}_{FRF} = \mathcal{L}_{trans} + \mathcal{L}_{recon}. \tag{4}$$

### 3.4 Hierarchical Mutual Information Maximization

The underlying assumption of knowledge distillation is that layers in the pre-trained teacher network can represent certain attributes of given inputs that exist in the task [15]. For successful knowledge transfer, the student network must learn to incorporate such attributes into its own learning. Nevertheless, previous studies [17, 38, 21] based on knowledge distillation simply constrain the consistency between the features of both networks and lack consideration of the intrinsic semantics and inherent properties of the features, leading to semantic misalignment. From the perspective of information theory [1], semantic alignment and attribute mining of representations can be characterized as maintaining high mutual information among the layers of the teacher and student networks. We construct a Hierarchical Mutual Information (HMI) maximization mechanism to implement sufficient semantic alignment and maximize mutual information. The core idea is to progressively align the semantics of representations through a hierarchical learning paradigm.

Specifically, the sentiment-relevant and modality-specific representations $\boldsymbol{Q}_m$ and $\boldsymbol{U}_m$ of all modalities for teacher and student networks are concatenated to obtain the sequences $[\boldsymbol{C}_L, \boldsymbol{C}_A, \boldsymbol{C}_V]$ and $[\boldsymbol{C}_L', \boldsymbol{C}_A', \boldsymbol{C}_V']$. Each element of the sequences is concatenated to yield the joint multimodal representations $\boldsymbol{H}^t$ and $\boldsymbol{H}^s$. The fully-connected layers are utilized to refine the representation $\boldsymbol{H}^w \in \mathbb{R}^{3d}$ with $w \in \{t, s\}$, yielding $\tilde{\boldsymbol{H}}^w \in \mathbb{R}^{3d}$. Moreover, we obtain the intermediate multi-scale representations of all layers, denoted as $\boldsymbol{I}_1^w \in \mathbb{R}^{2d}$, $\boldsymbol{I}_2^w \in \mathbb{R}^d$, and $\boldsymbol{I}_3^w \in \mathbb{R}^{2d}$. For the above five representations, we concatenate features of the same scale to obtain multi-scale representations $\boldsymbol{E}_1^w \in \mathbb{R}^{3d}$, $\boldsymbol{E}_2^w \in \mathbb{R}^{2d}$, and $\boldsymbol{E}_3^w \in \mathbb{R}^d$, which are utilized in the subsequent computation.

To estimate and compute the mutual information between representations, we define two random variables $\boldsymbol{X}$ and $\boldsymbol{Y}$. The $P(\boldsymbol{X})$ and $P(\boldsymbol{Y})$ are the marginal probability density function of $\boldsymbol{X}$ and $\boldsymbol{Y}$. The joint probability density function of $\boldsymbol{X}$ and $\boldsymbol{Y}$ is denoted as $P(\boldsymbol{X}, \boldsymbol{Y})$. The mutual information of the random variables $\boldsymbol{X}$ and $\boldsymbol{Y}$ is represented as:

$$I(\boldsymbol{X}; \boldsymbol{Y}) = \mathbb{E}_{p(\boldsymbol{x}, \boldsymbol{y})} \left[ \log \frac{p(\boldsymbol{x}, \boldsymbol{y})}{p(\boldsymbol{x})p(\boldsymbol{y})} \right]. \tag{5}$$

We only need to obtain the maximum value of the mutual information, without focusing on its exact value. Referring to Deep InfoMax [16], we estimate the mutual information between variables based on the Jensen-Shannon Divergence (JSD). The mutual information maximization issue translates into minimizing the JSD between the joint distribution $p(\boldsymbol{x}, \boldsymbol{y})$ and the marginal distribution $p(\boldsymbol{x})p(\boldsymbol{y})$.

$$JSD(p(\boldsymbol{x}, \boldsymbol{y}) \| p(\boldsymbol{x})p(\boldsymbol{y})) = \frac{1}{2} \left( D_{KL}(p(\boldsymbol{x}, \boldsymbol{y}) \| m) + D_{KL}(p(\boldsymbol{x})p(\boldsymbol{y}) \| m) \right), \tag{6}$$

where $m = \frac{1}{2}(p(\boldsymbol{x}, \boldsymbol{y}) + p(\boldsymbol{x})p(\boldsymbol{y}))$ and $D_{KL}$ is Kullback-Leibler divergence. Mutual information maximization is achieved by maximizing the dyadic lower bound of JSD, denoted as:

$$MI(\boldsymbol{x}, \boldsymbol{y}) = \mathbb{E}_{P(\boldsymbol{x}, \boldsymbol{y})}[-sp(-\mathcal{T}_\theta(\boldsymbol{x}, \boldsymbol{y})] + \mathbb{E}_{P(\boldsymbol{x})P(\boldsymbol{y})}[-sp(\mathcal{T}_\theta(\boldsymbol{x}, \boldsymbol{y})], \tag{7}$$

where $sp(w) = \log(1 + e^w)$ and $\mathcal{T}_\theta(\boldsymbol{x}, \boldsymbol{y})$ is the statistics network which is a neural network with parameters $\theta$. HMI maximizes the mutual information between hierarchical representations in knowledge distillation, whose optimization loss is expressed as:

$$\mathcal{L}_{HMI} = -\sum_{i=1}^{3} MI(\boldsymbol{E}_i^t, \boldsymbol{E}_i^s). \tag{8}$$

### 3.5 Hierarchical Adversarial Learning

Considering that the teacher network has more robust and stable representation distributions, we also need to encourage the alignment of representation distributions in the latent space. Traditional methods [38, 17, 21] simply minimize the KL divergence between both networks, which easily disturbs the underlying learning of the student network in the deep layers, leading to confounded distributions and unrobust joint multimodal representations.

To this end, we propose a Hierarchical Adversarial Learning (HAL) mechanism for incrementally aligning the latent distributions between representations of student and teacher networks. The central principle is that the student network tries to generate representations to mislead the discriminator $\mathcal{D}_e(\cdot)$, while $\mathcal{D}_e(\cdot)$ discriminates between the representations of the student and teacher networks. In practice, $\mathcal{D}_e(\cdot)$ is the fully-connected layers. Specifically, given multi-scale representations of $\boldsymbol{E}_1^w \in \mathbb{R}^{3d}$, $\boldsymbol{E}_2^w \in \mathbb{R}^{2d}$, and $\boldsymbol{E}_3^w \in \mathbb{R}^d$ with $w \in \{t, s\}$, we implement adversarial learning on the same-scale representations of the teacher and student networks to hierarchically supervise consistency. The objective function of HAL is formatted as:

$$\mathcal{L}_{HAL} = \sum_{i=1}^{3} \log(1 - \mathcal{D}_e(\boldsymbol{E}_i^s)) + \log(\mathcal{D}_e(\boldsymbol{E}_i^t)). \tag{9}$$

### 3.6 Optimization Objectives

The $\tilde{\boldsymbol{H}}^t$ and $\tilde{\boldsymbol{H}}^s$ of the teacher and student networks are fed into their task-specific classifiers to produce logits $\boldsymbol{L}^t$ and $\boldsymbol{L}^s$, respectively, and the consistency of both is constrained with KL divergence loss, denoted as $\mathcal{L}_{KL} = KL(\boldsymbol{L}^t, \boldsymbol{L}^s)$. The $\boldsymbol{L}^s$ is used for sentiment recognition and supervised with task loss, represented as $\mathcal{L}_{task}$. For the classification and regression tasks, we use cross-entropy and MSE loss as the task losses, respectively. The overall training objective $\mathcal{L}_{total}$ is expressed as $\mathcal{L}_{total} = \mathcal{L}_{task} + \mathcal{L}_{FRF} + \mathcal{L}_{HMI} + \mathcal{L}_{HAL} + \mathcal{L}_{KL}$.

## 4 Experiments

### 4.1 Datasets and Evaluation Metrics

We conduct our experiments on three MSA benchmarks, including MOSI [71], MOSEI [72], and IEMOCAP [4]. The experiments are performed under the word-aligned setting. MOSI is a realistic dataset for MSA. It comprises 2,199 short monologue video clips taken from 93 YouTube movie review videos. There are 1,284, 229, and 686 video clips in train, valid, and test data, respectively. MOSEI is a dataset consisting of 22,856 movie review video clips, which has 16,326, 1,871, and 4,659 samples in train, valid, and test data. Each sample of MOSI and MOSEI is labelled by human annotators with a sentiment score of -3 (strongly negative) to +3 (strongly positive). On the MOSI and MOSEI datasets, we utilize two evaluation metrics, including the Mean Absolute Error (MAE) and F1 score computed for positive/negative classification results. The IEMOCAP dataset consists of 4,453 samples of video clips. Its predetermined data partition has 2,717, 798, and 938 samples in train, valid, and test data. As recommended by [48], four emotions (*i.e.,* happy, sad, angry, and neutral) are selected for emotion recognition. The F1 score is used as the metric.

### 4.2 Implementation Details

**Feature Extraction.** The Glove embedding [36] is used to convert the video transcripts to obtain a 300-dimensional vector for the language modality. For the audio modality, we employ the COVAREP

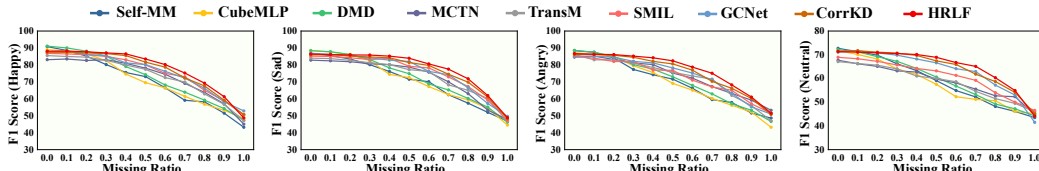

Figure 3: Comparison results of intra-modality missingness on IEMOCAP. We report on the F1 score metric for the happy, sad, angry, and neutral categories.

toolkit [7] to extract 74-dimensional acoustic features, including 12 Mel-frequency cepstral coefficients, voiced/unvoiced segmenting features, and glottal source parameters. For the visual modality, we utilize the Facet [2] to indicate 35 facial action units that record facial movement.

**Experimental Setup.** Regarding the MOSI [71] and MOSEI [72] datasets, we use the aligned multimodal sequences therein (*e.g.*, all sequences of modalities have length 300) as the original input for the HRLF. All models are built on the Pytorch [34] toolbox with four NVIDIA Tesla V100 GPUs. The Adam optimizer [20] is employed for network optimization. For MOSI, MOSEI, and IEMOCAP, the detailed hyper-parameter settings are as follows: the learning rates are $\{1e-3, 2e-3, 4e-3\}$, the batch sizes are $\{128, 16, 32\}$, the epoch numbers are $\{50, 20, 30\}$, and the attention heads are $\{10, 8, 10\}$. The embedding dimension is $40$ on all three datasets. The raw features at the modality missing positions are replaced by zero vectors. For a fair comparison, we re-implement the State-Of-The-Art (SOTA) methods and combine them with our experimental paradigms. All experimental results are averaged over multiple experiments using five different random seeds.

### 4.3 Comparison with State-of-the-art Methods

We conduct a comparison between HRLF and eight representative, reproducible state-of-the-art (SOTA) methods, including complete-modality methods: Self-MM [66], CubeMLP [42], and DMD [25], and missing-modality methods: 1) joint learning methods (*i.e.*, MCTN [37], TransM [50], and CorrKD [24]), and 2) generative methods (*i.e.*, SMIL [29] and GCNet [26]). The extensive experiments are designed to comprehensively assess the robustness and effectiveness of HRLF in scenarios involving both intra-modality and inter-modality missingness.

**Robustness to Intra-modality Missingness.** We simulate intra-modality missingness by randomly discarding frame-level features in sequences with ratio $p \in \{0.1, 0.2, \cdots, 1.0\}$. To visualize the robustness of all models, Figure 3 and 4 show the performance curves of the models for different ratios $p$. We have the following important observations. (i) As the ratio $p$ increases, the performance of all models declines. This phenomenon demonstrates that intra-modality missingness leads to significant sentiment semantic loss and fragile multimodal representations. (ii) Compared to complete-modality methods, our HRLF demonstrates notable performance advantages in missing-modality testing conditions and competitive performance in complete-modality testing conditions. This is because complete-modality methods rely on the assumption of data completeness, while training paradigms for missing modalities excel in capturing and reconstructing valuable sentiment semantics from incomplete multimodal data. (iii) In contrast to the missing-modality methods, our HRLF demonstrates the highest

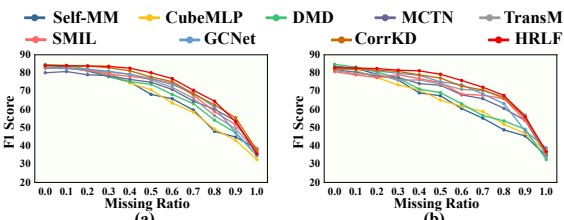

Figure 4: Comparison results of intra-modality missingness on (a) MOSI and (b) MOSEI.

level of robustness. Through the purification of sentiment semantics and the dual alignment of representations, the student network masters the core competencies of precisely reconstructing missing semantics and generating robust multimodal representations.

**Robustness to Inter-modality Missingness.** To simulate the case of inter-modality missingness, we remove certain entire modalities from the samples. Tables 1 and 2 contrast the models' resilience to inter-modality missingness. The notation "$\{l\}$" signifies that only the language modality is available,

Table 1: Comparison of performance under six possible testing conditions of inter-modality missingness and the complete-modality testing condition on the MOSI and MOSEI datasets. T-test is conducted on "Avg." column. ∗ indicates that $p < 0.05$ (compared with the SOTA CorrKD).

| Datasets | Models | Testing Conditions | | | | | | | |
|---|---|---|---|---|---|---|---|---|---|
| | | $\{l\}$ | $\{a\}$ | $\{v\}$ | $\{l,a\}$ | $\{l,v\}$ | $\{a,v\}$ | Avg. | $\{l,a,v\}$ |
| MOSI | Self-MM [66] | 67.80 | 40.95 | 38.52 | 69.81 | 74.97 | 47.12 | 56.53 | **84.64** |
| | CubeMLP [42] | 64.15 | 38.91 | 43.24 | 63.76 | 65.12 | 47.92 | 53.85 | 84.57 |
| | DMD [25] | 68.97 | 43.33 | 42.26 | 70.51 | 68.45 | 50.47 | 57.33 | 84.50 |
| | MCTN [37] | 75.21 | 59.25 | 58.57 | 77.81 | 74.82 | 64.21 | 68.31 | 80.12 |
| | TransM [50] | 77.64 | 63.57 | 56.48 | 82.07 | 80.90 | 67.24 | 71.32 | 82.57 |
| | SMIL [29] | 78.26 | 67.69 | 59.67 | 79.82 | 79.15 | 71.24 | 72.64 | 82.85 |
| | GCNet [26] | 80.91 | 65.07 | 58.70 | **84.73** | 83.58 | 70.02 | 73.84 | 83.20 |
| | CorrKD [24] | 81.20 | 66.52 | 60.72 | 83.56 | 82.41 | 73.74 | 74.69 | 83.94 |
| | **HRLF (Ours)** | **83.36** | **69.47** | **64.59** | 83.82 | 83.56 | **75.62** | **76.74**∗ | 84.15 |
| MOSEI | Self-MM [66] | 71.53 | 43.57 | 37.61 | 75.91 | 74.62 | 49.52 | 58.79 | 83.69 |
| | CubeMLP [42] | 67.52 | 39.54 | 32.58 | 71.69 | 70.06 | 48.54 | 54.99 | 83.17 |
| | DMD [25] | 70.26 | 46.18 | 39.84 | 74.78 | 72.45 | 52.70 | 59.37 | **84.78** |
| | MCTN [37] | 75.50 | 62.72 | 59.46 | 76.64 | 77.13 | 64.84 | 69.38 | 81.75 |
| | TransM [50] | 77.98 | 63.68 | 58.67 | 80.46 | 78.61 | 62.24 | 70.27 | 81.48 |
| | SMIL [29] | 76.57 | 65.96 | 60.57 | 77.68 | 76.24 | 66.87 | 70.65 | 80.74 |
| | GCNet [26] | 80.52 | 66.54 | 61.83 | 81.96 | 81.15 | 69.21 | 73.54 | 82.35 |
| | CorrKD [24] | 80.76 | 66.09 | 62.30 | 81.74 | **81.28** | 71.92 | 74.02 | 82.16 |
| | **HRLF (Ours)** | **82.05** | **69.32** | **64.90** | **82.62** | 81.09 | **73.80** | **75.63**∗ | 82.93 |

while the audio and visual modalities are missing. "$\{l,a,v\}$" denotes the complete-modality testing condition where all modalities are available. "Avg." indicates the average performance across six missing-modality testing conditions. We have the following key findings: **(i)** The inter-modality missingness leads to a decline in performance for all models, indicating that integrating complementary information from diverse modalities enhances the sentiment semantics within joint representations. **(ii)** Across all six testing conditions involving inter-modality missingness, our HRLF consistently demonstrates superior performance among the majority of metrics, affirming its robustness. For example, on the MOSI dataset, HRLF's average F1 score is improved by $2.05\%$ compared to CorrKD, and in particular by $3.87\%$ in the testing condition where only visual modality is available (*i.e.*, $\{v\}$). The advantage comes from its learning of fine-grained representation factorization and the hierarchical semantic alignment and distributional alignment. **(iii)** In unimodal testing scenarios, HRLF's performance using only the language modality significantly exceeds other configurations, showing performance similar to that of the complete-modality setup. In bimodal testing scenarios, configurations involving the language modality exhibit superior performance, even outperforming the complete-modality setup in specific metrics. This phenomenon underscores the richness of sentiment semantics within the language modality and its dominance in sentiment inference and missing semantic reconstruction processes.

## 4.4 Ablation Studies

To affirm the effectiveness and indispensability of the module and mechanisms and strategies proposed in HRLF, we perform ablation experiments under two missing-modality scenarios on the MOSI dataset, as shown in Table 3 and Figure 5. We have the following important observations: **(i)** First, when the FRF is removed, sentiment-relevant and modality-specific information in the modalities are confused, hindering sentiment recognition and leading to significant performance degradation. This phenomenon demonstrates the effectiveness of the proposed representation factorization paradigm for adequate capture of valuable sentiment semantics. **(ii)** When our HMI

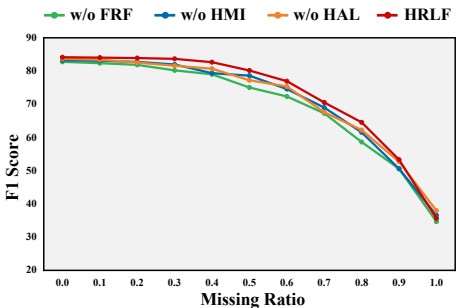

Figure 5: Ablation results of intra-modality missingness case on the MOSI dataset.

Table 2: Comparison of performance under six possible testing conditions of inter-modality missingness and the complete-modality testing condition on the IEMOCAP dataset. T-test is conducted on "Avg." column. $*$ indicates that $p < 0.05$ (compared with the SOTA CorrKD).

| Models | Metrics | Testing Conditions | | | | | | | |
|---|---|---|---|---|---|---|---|---|---|
| | | $\{l\}$ | $\{a\}$ | $\{v\}$ | $\{l,a\}$ | $\{l,v\}$ | $\{a,v\}$ | Avg. | $\{l,a,v\}$ |
| Self-MM [66] | Happy | 66.9 | 52.2 | 50.1 | 69.9 | 68.3 | 56.3 | 60.6 | 90.8 |
| | Sad | 68.7 | 51.9 | 54.8 | 71.3 | 69.5 | 57.5 | 62.3 | 86.7 |
| | Angry | 65.4 | 53.0 | 51.9 | 69.5 | 67.7 | 56.6 | 60.7 | 88.4 |
| | Neutral | 55.8 | 48.2 | 50.4 | 58.1 | 56.5 | 52.8 | 53.6 | **72.7** |
| CubeMLP [42] | Happy | 68.9 | 54.3 | 51.4 | 72.1 | 69.8 | 60.6 | 62.9 | 89.0 |
| | Sad | 65.3 | 54.8 | 53.2 | 70.3 | 68.7 | 58.1 | 61.7 | **88.5** |
| | Angry | 65.8 | 53.1 | 50.4 | 69.5 | 69.0 | 54.8 | 60.4 | 87.2 |
| | Neutral | 53.5 | 50.8 | 48.7 | 57.3 | 54.5 | 51.8 | 52.8 | 71.8 |
| DMD [25] | Happy | 69.5 | 55.4 | 51.9 | 73.2 | 70.3 | 61.3 | 63.6 | **91.1** |
| | Sad | 65.0 | 54.9 | 53.5 | 70.7 | 69.2 | 61.1 | 62.4 | 88.4 |
| | Angry | 64.8 | 53.7 | 51.2 | 70.8 | 69.9 | 57.2 | 61.3 | **88.6** |
| | Neutral | 54.0 | 51.2 | 48.0 | 56.9 | 55.6 | 53.4 | 53.2 | 72.2 |
| MCTN [37] | Happy | 76.9 | 63.4 | 60.8 | 79.6 | 77.6 | 66.9 | 70.9 | 83.1 |
| | Sad | 76.7 | 64.4 | 60.4 | 78.9 | 77.1 | 68.6 | 71.0 | 82.8 |
| | Angry | 77.1 | 61.0 | 56.7 | 81.6 | 80.4 | 58.9 | 69.3 | 84.6 |
| | Neutral | 60.1 | 51.9 | 50.4 | 64.7 | 62.4 | 54.9 | 57.4 | 67.7 |
| TransM [50] | Happy | 78.4 | 64.5 | 61.1 | 81.6 | 80.2 | 66.5 | 72.1 | 85.5 |
| | Sad | 79.5 | 63.2 | 58.9 | 82.4 | 80.5 | 64.4 | 71.5 | 84.0 |
| | Angry | 81.0 | 65.0 | 60.7 | 83.9 | 81.7 | 66.9 | 73.2 | 86.1 |
| | Neutral | 60.2 | 49.9 | 50.7 | 65.2 | 62.4 | 52.4 | 56.8 | 67.1 |
| SMIL [29] | Happy | 80.5 | 66.5 | 63.8 | 83.1 | 81.8 | 68.2 | 74.0 | 86.8 |
| | Sad | 78.9 | 65.2 | 62.2 | 82.4 | 79.6 | 68.2 | 72.8 | 85.2 |
| | Angry | 79.6 | 67.2 | 61.8 | 83.1 | 82.0 | 67.8 | 73.6 | 84.9 |
| | Neutral | 60.2 | 50.4 | 48.8 | 65.4 | 62.2 | 52.6 | 56.6 | 68.9 |
| GCNet [26] | Happy | 81.9 | 67.3 | 66.6 | 83.7 | 82.5 | 69.8 | 75.3 | 87.7 |
| | Sad | 80.5 | 69.4 | 66.1 | 83.8 | 81.9 | 70.4 | 75.4 | 86.9 |
| | Angry | 80.1 | 66.2 | 64.2 | 82.5 | 81.6 | 68.1 | 73.8 | 85.2 |
| | Neutral | 61.8 | 51.1 | 49.6 | 66.2 | 63.5 | 53.3 | 57.6 | 71.1 |
| CorrKD [24] | Happy | 82.6 | 69.6 | 68.0 | 84.1 | 82.0 | 70.0 | 76.1 | 87.5 |
| | Sad | 82.7 | **71.3** | 67.6 | 83.4 | 82.2 | 72.5 | 76.6 | 85.9 |
| | Angry | 82.2 | 67.0 | 65.8 | 83.9 | 82.8 | 67.3 | 74.8 | 86.1 |
| | Neutral | 63.1 | 54.2 | 52.3 | 68.5 | 64.3 | **57.2** | 59.9 | 71.5 |
| **HRLF (Ours)** | Happy | **84.9** | **71.8** | **69.7** | **86.4** | **85.6** | **72.3** | **78.5**$^*$ | 88.1 |
| | Sad | **83.7** | 71.1 | **69.0** | **85.3** | **83.9** | **73.6** | **77.8**$^*$ | 86.4 |
| | Angry | **83.4** | **69.1** | **67.2** | **84.5** | **83.5** | **70.9** | **76.4**$^*$ | 86.7 |
| | Neutral | **66.8** | **56.1** | **54.5** | **68.9** | **67.0** | 56.9 | **61.7**$^*$ | 71.3 |

is eliminated, the worse performance demonstrates that aligning the high-level semantics in the representation by maximizing mutual information can generate favorable joint representations for the student network. **(iii)** Finally, we remove HAL, and the declined results illustrate that multi-scale adversarial learning can effectively align the representation distributions of student and teacher networks, thus effectively constraining the consistency across representations. This paradigm facilitates the recovery of missing semantics.

### 4.5 Qualitative Analysis

To intuitively show the robustness of the proposed framework against modality missingness, we randomly select 100 samples in each emotion category on the IEMOCAP testing set to perform the visualization evaluation. The comparison models include CubeMLP [42] (complete-modality method), TransM [50] (joint learning-based missing-modality method), and GCNet [26] (generation-based missing-modality method). **(i)** As shown in Figure 6, CubeMLP fails to cope with the missing modality challenge because representations with different emotion categories are heavily confounded,

Table 3: Ablation results of inter-modality missingness case on the MOSI dataset.

| Models | Testing Conditions | | | | | | | |
|---|---|---|---|---|---|---|---|---|
| | $\{l\}$ | $\{a\}$ | $\{v\}$ | $\{l,a\}$ | $\{l,v\}$ | $\{a,v\}$ | Avg. | $\{l,a,v\}$ |
| **HRLF (Full)** | **83.36** | **69.47** | **64.59** | **83.82** | **83.56** | **75.62** | **76.74** | **84.15** |
| w/o FRF | 80.85 | 67.06 | 61.78 | 81.94 | 81.38 | 73.58 | 74.43 | 82.76 |
| w/o HMI | 81.54 | 67.72 | 62.70 | 82.45 | 81.90 | 74.22 | 75.09 | 83.25 |
| w/o HAL | 82.03 | 68.09 | 63.11 | 83.12 | 82.67 | 74.59 | 75.60 | 83.67 |

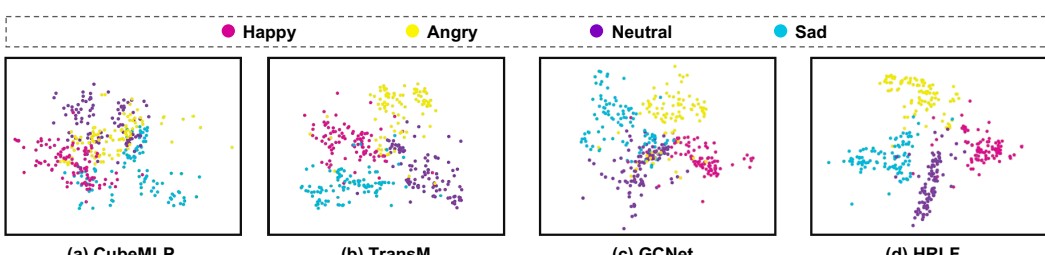

| | | | |
|---|---|---|---|
| **(a) CubeMLP** | **(b) TransM** | **(c) GCNet** | **(d) HRLF** |

Figure 6: Visualization of representations from different methods with four emotion categories on the IEMOCAP testing set. The default testing conditions contain intra-modality missingness (*i.e.*, missing rate $p = 0.5$ ) and inter-modality missingness (*i.e.*, only the language modality is available).

leading to the worst results. **(ii)** Although TransM and GCNet mitigate the indistinguishable emotion semantics to some extent, their performance is sub-optimal since the distribution boundaries of the different emotion representations are generally ambiguous and coupled. **(iii)** In comparison, our HRLF enables representations belonging to the same emotion category to form compact clusters, while representations of different categories are well separated. The above phenomenon benefits from the effective extraction of sentiment semantics and the precise filtering of task redundant information by the proposed hierarchical representation learning framework, which results in better joint multimodal representations. This further confirms the robustness and superiority of our framework.

## 5 Conclusion and Discussion

In this paper, we present a Hierarchical Representation Learning Framework (HRLF) to address diverse missing modality dilemmas in the MSA task. Specifically, we mine sentiment-relevant representations through a fine-grained representation factorization module. Additionally, the hierarchical mutual information maximization mechanism and the hierarchical adversarial learning mechanism are proposed for semantic and distributional alignment of representations of student and teacher networks to accurately reconstruct missing semantics and produce robust joint multimodal representations. Comprehensive experiments validate the superiority of our framework.

**Discussion of Limitation and Future Work.** The current method defines the modality missing cases as both inter-modality missingness and intra-modality missingness. Nevertheless, in real-world applications, modality missing cases may be very intricate and difficult to simulate. Consequently, the proposed method may suffer some minor performance loss when applied to real-world scenarios. In the future, we will explore more intricate modality missing cases and design suitable algorithms to compensate for this deficiency.

**Discussion of Broad Impacts.** The positive impact of our approach lies in the ability to significantly improve the robustness and stability of multimodal sentiment analysis systems against heterogeneous modality missingness in real-world applications. Nevertheless, this technology may have a negative impact when it falls into the wrong hands, *e.g.*, the proposed model is used for malicious purposes by injecting biased priors to recognize the emotions of specific groups.

## 6 Acknowledgements

This work was supported in part by National Key R&D Program of China 2021ZD0113502 and in part by Shanghai Municipal Science and Technology Major Project 2021SHZDZX0103.

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
