# OpenReview forum: "Toward Robust Incomplete Multimodal Sentiment Analysis via Hierarchical Representation Learning"
_NeurIPS.cc/2024/Conference — NeurIPS 2024 poster_

### Official Review · Reviewer_Zh5E · 2024-07-04

**Soundness:** 3
**Presentation:** 1
**Contribution:** 2
**Rating:** 5
**Confidence:** 4

**Summary:**

The paper addresses the issue of missing modality information in multimodal systems and proposes solutions for two key problems:
1. Excessively complex feature interactions lead to information redundancy and cumulative errors.
2. Previous works did not align representations semantically.

In enhancing multimodal representations, the core approach utilizes the transformer’s encoder and decoder. It strengthens inter-modal interactions post-encoder to ensure semantic relevance is maintained in the decoder's reconstruction.
The model achieves state-of-the-art (SOTA) performance and includes detailed ablation studies and experimental analyses.

**Strengths:**

1. The model considers the differences between semantic and modality representations, utilizing the transformer’s encoder and decoder processes for reconstruction to facilitate inter-modal interactions.
2. It accounts for the impact of Mutual Information (MI) on the learned representations, using MI to enhance the quality of these representations.
3. The paper conducts thorough experiments, addressing the model's robustness when adjusting the modality missing ratio.

**Weaknesses:**

1. The use of numerous symbols in both the figures and text increases the complexity and time required for understanding. DIFFICULT TO READ
2. In the introduction, the problem is not clearly stated; it is too brief. It only specifically mentions SMIL's approach without thoroughly analyzing the issues present in previous methods.
3. It appears that the model relies heavily on MRM (Modality Stochastic Missing) for masking certain information to generate task-specific data. This raises concerns about the model's dependency on MRM. If MRM focuses on masking emotional words, does it hinder the model's understanding?
4. The paper does not consider using large models to address this problem, nor does it compare the performance with that of large models.
5. The HMI (Hierarchical Mutual Information) module applies Bengio's MI concepts rather straightforwardly, lacking in innovation.
6. HAL (Hierarchical Adversarial Learning), presented as a separate contribution, does not make a significant impact in terms of performance or design. The hierarchical aspect merely reflects multi-scale representation.

**Questions:**

Application Scenario: In the introductory example, if the crucial information "bored" is missing in L (Linguistic), A (Acoustic), and V (Visual) modalities, how is it still relevant to determine the task? What basis is used for this determination?
Model Design: Why is it necessary to include a modality encoder in the model? Ensuring semantic interactions across different modalities should be sufficient.
Did the baselines you compared against also use the SAME masking strategy as you did?

**Limitations:**

See Weaknesses and Questions.

---

> ### Author Rebuttal · Authors · 2024-08-07
>
> **Q1**: About symbolic representation in figures and texts.
>
> **A1**: Thank you for the reminder! It is necessary that we use symbols in figures and text to represent the data flow and workflow of each component in the framework. We will improve the representation in the revision to make it easier to read.
> ***
> **Q2**: About problems in the previous methods.
>
> **A2**:
> *  As stated in lines 34-38 in the main manuscript, we have summarized the common problems in previous methods:
>
>     * performing complex feature interactions on missing modalities leads to a large amount of information redundancy and cumulative errors, which affects the effectiveness of affective semantics extraction.
>
>     * The lack of consideration of semantic alignment and distributional alignment during feature reconstruction results in the inability to accurately recover features and produce robust joint multimodal representations.
>
> * In this paper, we only take SMIL as an example, which also suffers from the above issues and is not a special case. We promise to improve the representation in the revision.
> ***
> **Q3**: If MRM focuses on masking emotional words, does it hinder the model's understanding?
>
> **A3**: Thank you for your comment! We need to clarify that:
> *  the MRM process is stochastic and does not mask out some specific words, and this stochastic paradigm effectively enhances the model's ability to deal with complex modality missing situations.
> * When MRM is discarded, the decreased performance of the model proves its indispensability and importance, as shown in Table. 4 in **reply.pdf**. The testing conditions include Intra-Modality Missingness (Intra-MM), Inter-Modality Missingness (Inter-MM), and Complete Modality (CM).
> ***
> **Q4**: Please compare the proposed framework with the larger models.
>
> **A4**: Insightful comments. We want to emphasize the current immaturity of using large models to solve MSA tasks with three modalities: language, audio, and video. We did our best to select VideoLLaMA-1/2 that has good support for all three modalities in Table 5 of **reply.pdf** for comparison experiments.The testing conditions include Intra-Modality Missingness (Intra-MM), Inter-Modality Missingness (Inter-MM), and Complete Modality (CM). We find it difficult for large models to deliver significant gains when modality semantics are incomplete. In contrast, large models show promising potential for traditional perception tasks when all modalities are available.
> ***
> **Q5**: Please describe the technical contributions of HMI and HAL.
>
> **A5**: We offer the following technical clarifications:
> * The concept of mutual information and adversarial learning are not simply combined and used, but are specifically designed to address the problems and limitations of existing MSA methods under uncertain missing modality.  Specifically, existing methods lack effective supervision of semantic alignment and distributional alignment during feature reconstruction, resulting in the inability to accurately restore missing sentiment semantics, as stated in lines 37-38 & 87-90 & 157-160 & 184-188 in the main manuscript. In contrast, the proposed interaction paradigm guides the reconstruction of missing sentiment information both at the semantic level and at the distributional level, thus recovering realistic modality features as much as possible.
> * HMI and HAL bring favorable performance gains, as shown in Table 3 in the main manuscript and Table 6 in the Appendix, which demonstrate the necessity and importance of both mechanisms.
> ***
> **Q6**: Application Scenario: In the introductory example, if the crucial information "bored" is missing in L (Linguistic), A (Acoustic), and V (Visual) modalities, how is it still relevant to determine the task? What basis is used for this determination? Model Design: Why is it necessary to include a modality encoder in the model? Ensuring semantic interactions across different modalities should be sufficient. Did the baselines you compared against also use the SAME masking strategy as you did?
>
> **A6**: Thank you for your comments!
>
> * Application scenario: When critical information "bored" from all three modalities is missing, the model is still able to give judgments based on the following two grounds:
>
> 1) The inherent imbalance in the label distribution in the MSA task (more positive than negative samples) leads the model to potentially rely on label bias as a statistical shortcut to perform predictions. Bias-driven predictions contain task-relevant a priori information to some extent [1].
>
> 2) For multimodal sequential data with temporal asynchrony, MSA is often able to capture global contextual dependencies among elements during temporal modeling to provide task-relevant contextual semantics.
>
> * Model design: modal encoders are used to unify dimensions and provide refined modal representations for subsequent semantic interactions. For a fair comparison, the baseline and our framework use the same mask strategy.
>
> [1] Yang, Dingkang, et al. "Towards multimodal sentiment analysis debiasing via bias purification." In ECCV 2024.

---

### Official Review · Reviewer_f4LC · 2024-07-06

**Soundness:** 4
**Presentation:** 4
**Contribution:** 3
**Rating:** 8
**Confidence:** 4

**Summary:**

The paper addresses the challenge of data incompleteness in Multimodal Sentiment Analysis (MSA). It introduces a novel approach called the Language-dominated Noise-resistant Learning Network (LNLN). The LNLN leverages the dense sentiment information in the language modality, considered the dominant modality, to improve robustness across various noise scenarios. It features two main components: a dominant modality correction (DMC) module and a dominant modality-based multimodal learning (DMML) module, which enhance the quality of the dominant modality representations. The model's performance was evaluated using datasets like MOSI, MOSEI, and SIMS, demonstrating superior robustness and accuracy compared to existing baselines. The comprehensive experiments provide new insights and a thorough comparative analysis in the context of incomplete data, advancing the field of MSA.

**Strengths:**

The introduction of the Language-dominated Noise-resistant Learning Network (LNLN) is innovative, addressing the issue of data incompleteness effectively by prioritizing the language modality, which is typically rich in sentiment information.

The authors conduct thorough experiments on well-known datasets (MOSI, MOSEI, SIMS), adding credibility to their findings. The detailed comparison with existing methods under diverse noise scenarios is particularly valuable.

The use of the Dominant Modality Correction (DMC) module and Dominant Modality Based Multimodal Learning (DMML) module is well-justified and systematically enhances the model’s robustness by ensuring the quality of dominant modality representations.

**Weaknesses:**

The focus on language as the dominant modality, while justified, may not generalize well to scenarios where other modalities (like visual or auditory) are equally or more critical. This could limit the applicability of the model to certain types of data or tasks.

**Questions:**

Extending the approach to consider scenarios where visual or auditory data might be dominant could improve the versatility and applicability of the model.

More detailed ablation studies on MOSEI and SIMS would provide deeper insights into the workings and benefits of the proposed model.

**Limitations:**

yes

---

> ### Author Rebuttal · Authors · 2024-08-07
>
> **Q1**: About model applicability in different scenarios.
>
> **A1**: Many thanks to the reviewer for the constructive suggestions. We would like to clarify several points.
> (1) In the MSA task, the language modality contains more refined and rich sentiment semantics than the other modalities, and thus language plays a dominant role in MSA. (2) The training paradigm designed in this framework can effectively capture the complementary sentiment semantics among heterogeneous modalities, which enhances the applicability and scalability of the proposed model under multiple data types and multiple tasks. (3) Limited by rebuttal time, we promise to extend the proposed approach to other modality-dominated scenarios to provide comprehensive insights and perspectives in future work.
> ***
> **Q2**: Add more detailed ablation studies.
>
> **A2**: Valuable Recommendations! The results of the ablation study of the proposed framework on the MOSEI and SIMS datasets are shown in Table. 3 in **reply.pdf**. We will add these ablation experiments to the revision.
> ***

---

### Official Review · Reviewer_JfYH · 2024-07-06

**Soundness:** 3
**Presentation:** 3
**Contribution:** 3
**Rating:** 8
**Confidence:** 3

**Summary:**

The paper presents the Representation Factorization and Alignment (ReFA) framework for Multimodal Sentiment Analysis (MSA) under uncertain missing modalities. ReFA employs a fine-grained representation factorization module to extract sentiment-relevant and modality-specific representations through crossmodal translation and sentiment semantic reconstruction. It introduces a hierarchical mutual information maximization mechanism to align and reconstruct high-level semantics incrementally. Additionally, a hierarchical adversarial learning mechanism progressively aligns latent distributions to create robust joint multimodal representations. Experiments on three datasets show that ReFA significantly enhances MSA performance under both uncertain missing-modality and complete-modality conditions.

**Strengths:**

Strength:

1. One of the challenges in multimodal sentiment analysis is the potential for missing modality information in real-world scenarios. This study addresses this practical issue by proposing an effective algorithm with notable real-world applicability.

2. The motivation for the research is clearly articulated, pinpointing the shortcomings of existing studies with strong logical coherence.

3. The proposed algorithm achieves state-of-the-art (SOTA) results across relevant datasets, which validates its effectiveness to a significant extent.

**Weaknesses:**

Weakness:

1. The idea and design of Intra- and Inter-modality Translations are sound; however, the implementation of the translation loss is overly simplistic and lacks a task-specific approach, making the methodology appear somewhat naive.

2. Similarly, the Sentiment Semantic Reconstruction section suffers from the same issue, with a basic and unrefined approach that fails to leverage the complexity of the task.

3. Both sections give the impression that while Translations and Reconstruction are being performed, the methods are indistinguishable aside from their goals. This indicates a lack of differentiation in handling the unique characteristics of each type of information.

4. The HMI and HAL components seem to merely apply two loss functions to the multi-scale features of the teacher-student network. This approach is quite common in knowledge distillation and thus lacks significant innovation.

5. Moreover, the paper lacks relevant case studies to validate the effectiveness of the proposed algorithm. There is also an absence of error analysis to identify the limitations and shortcomings of the method.

**Questions:**

1. The idea and design of Intra- and Inter-modality Translations are sound; however, the implementation of the translation loss is overly simplistic and lacks a task-specific approach, making the methodology appear somewhat naive.

2. Similarly, the Sentiment Semantic Reconstruction section suffers from the same issue, with a basic and unrefined approach that fails to leverage the complexity of the task.

3. Both sections give the impression that while Translations and Reconstruction are being performed, the methods are indistinguishable aside from their goals. This indicates a lack of differentiation in handling the unique characteristics of each type of information.

4. The HMI and HAL components seem to merely apply two loss functions to the multi-scale features of the teacher-student network. This approach is quite common in knowledge distillation and thus lacks significant innovation.

5. Moreover, the paper lacks relevant case studies to validate the effectiveness of the proposed algorithm. There is also an absence of error analysis to identify the limitations and shortcomings of the method.

**Limitations:**

The current distillation framework relies on a teacher network trained with complete modality data. A potential limitation of this approach is that its applicability to scenarios with missing modalities is inherently constrained by the performance ceiling of the teacher network. This dependency may limit the effectiveness and generalizability of the framework in handling diverse cases of missing modality data.

---

> ### Author Rebuttal · Authors · 2024-08-07
>
> **Q1**: About modality translation and sentiment semantic reconstruction.
>
> **A1**:
> * Both modality translation and sentiment semantic reconstruction are designed for MSA tasks. Specifically, the core idea of modality translation is to utilize transitions among different modalities to achieve an effective extraction of sentiment-relevant representations. The purpose of the sentiment semantic reconstruction process is to ensure that the translation produces representations that can still contain sentiment semantics consistent with the original.
> * The design of translation loss and reconstruction loss is simple but effective, and the complex loss is difficult to train and performs poorly, the experimental results are shown in Table. 2 in **reply.pdf**. The testing conditions include Intra-Modality Missingness (Intra-MM), Inter-Modality Missingness (Inter-MM), and Complete Modality (CM). Obviously, the design in this paper has the best performance. They center on the form of supervision rather than the loss function.
> ***
>
> **Q2**: About the technical contribution of HMI and HAL.
>
> **A2**: Thank you for your comments! We need to clarify that:
> * The multiscale feature-based knowledge distillation paradigm is indeed general and effective. However, the focus of this paper is not on the innovation of the knowledge distillation framework, but on fully utilizing the effective multiscale supervision paradigm in knowledge distillation to achieve hierarchical constraints between representations and missing feature reconstruction.
> * Distinguishing from the traditional simple constraint approach based on multi-scale features (\emph{e.g.}, L2 distance), we propose a joint alignment mechanism based on mutual information maximization at the semantic level and adversarial learning at the distributional level.
> * Instead of simply employing HMI and HAL, we address the lack of effective alignment of semantics and distributions in the feature reconstruction process for existing MSA methods under the missing modality cases, as stated in lines 37-38 & 87-90 & 157-160 & 184-188 in the main manuscript. The ablation experiments in Section 4.4 in the main manuscript and Section A.3 in the Appendix demonstrate the superiority of the proposed method.
> ***
>
> **Q3**: About the case studies and error analysis.
>
> **A3**: Valuable suggestions!
> * In order to better demonstrate the effectiveness of the proposed method, we used two challenging cases for case studies as shown in Fig. 3 in **reply.pdf**, where the underlined blue words may express emotional polarity and the missing modality is marked with a red dotted line.From the figure, we can find:  1) In E1, all models generate correct results though the visual modality is missing. Due to the strong guidance of the textual word  "amazing", the positive This case reveals that the conventional approaches can be well-performed when existing modalities express the same explicit semantics.  2) In E2, the textual modality expresses positive polarity, while the visual modality tends to be negative because of the frown and close lips on facial features. It is really hard to determine the polarity when the acoustic modality is missing. Specifically, Self-MM and CubeMLP misclassify the emotion as negative. Specifically, Self-MM and CubeMLP misclassify the emotion as negative, and the other approaches except ReFA all predict positive sentiment in terms of the dominance of the language modality. modality. In contrast, our framework recognizes correctly. This advantage stems from the present framework's factorization and capture of sentiment- relevant semantics, as well as the hierarchical knowledge distillation module's alignment of semantics and distributions, which accurately reconstructs missing features and produces robust joint multimodal representations.
> * We have conducted T-tests in Table. 1 and Table. 2 in the main manuscript, and the stable and highly significant experimental results demonstrate the superiority of the proposed method. Furthermore, we have described the limitations of our framework, as stated in the Section A.6 in the Appendix.
> ***
> **Q4**: About the generalization of the framework using the teacher network.
>
> **A4**:  Valuable insights!
> * We have clarified this limitation, as stated in Section A.6 in the Appendix. In the future, we will strive to optimize the generality of the approach, e.g., by using the self-distillation paradigm in the framework.
> * The teacher network is trained on the original complete-modality data, which serves as a high-quality reference that transfers holistic knowledge contained in the complete samples to the student network. The student network accurately recovers the missing semantics during semantic and distributional alignment to the teacher network.
> * The teacher network covers the information of the complete modality, and this supervised paradigm is effective and generalizable in a variety of modality missing scenarios, thus enhancing the generalization of the student network.

---

> > ### Comment · Reviewer_JfYH · 2024-08-09
> >
> > Thank you for your response and thorough explanations. I have revised my rating to 8.

---

> > > ### Author Response · Authors · 2024-08-09
> > > **Response to Reviewer JfYH**
> > >
> > > We thank the reviewer for the meticulous advice!

---

### Official Review · Reviewer_zmpp · 2024-07-06

**Soundness:** 3
**Presentation:** 3
**Contribution:** 4
**Rating:** 7
**Confidence:** 4

**Summary:**

The paper addresses the challenges of multimodal sentiment analysis (MSA) in real-world applications, particularly when some modalities may be missing, which can hinder the effectiveness of the analysis. The authors propose a framework called Representation Factorization and Alignment (ReFA) to tackle the issue of uncertain missing modalities in MSA.

The ReFA framework consists of three core components:

1. Fine-grained Representation Factorization (FRF) module: This module extracts valuable sentiment information by factorizing each modality into sentiment-relevant and modality-specific representations through cross-modal translation and sentiment semantic reconstruction.

2. Hierarchical Mutual Information (HMI) maximization mechanism: This mechanism incrementally maximizes the mutual information between multi-scale representations to align and reconstruct the high-level semantics in the representations.

3. Hierarchical Adversarial Learning (HAL) mechanism: This mechanism progressively aligns and adapts the latent distributions of the representations to produce robust joint multimodal representations.

The authors conducted comprehensive experiments on three datasets, demonstrating that the ReFA framework significantly improves MSA performance under both uncertain missing-modality and complete-modality testing conditions.

**Strengths:**

Originality:
1. The paper proposes a Representation Factorization and Alignment (ReFA) framework to address multimodal sentiment analysis under uncertain missing modalities.
2. Introduces innovative components like fine-grained representation factorization, hierarchical mutual information maximization, and hierarchical adversarial learning.

Quality:
1. Comprehensive experiments on three datasets (MOSI, MOSEI, IEMOCAP) demonstrate significant performance improvements.
2. Ablation studies validate the effectiveness of each proposed component.
3. Qualitative analysis with visualizations provides intuitive understanding of the framework's robustness.

Clarity:
1. The paper is well-structured, with clear sections on related work, methodology, and experiments.
2. Figures and tables effectively illustrate the framework and results.

Significance:
1. Addresses an important real-world challenge of missing modalities in multimodal sentiment analysis.
2. Shows consistent performance improvements over state-of-the-art methods across different missing modality scenarios.
3. The framework's robustness to both intra-modality and inter-modality missingness enhances its practical applicability.

**Weaknesses:**

1. The paper lacks a detailed discussion on the computational complexity and runtime performance of the proposed framework compared to existing methods.
2. While the proposed ReFA framework is innovative, the individual components (such as mutual information maximization and adversarial learning) have been explored in other contexts. The novelty primarily lies in their specific combination and application to MSA with missing modalities.
3. The paper did not mention the models that were used for the final classification or regression, only mentioned feature extraction models.
4. The paper doesn't discuss potential limitations of the approach or cases where it might not perform well.
5. There's no discussion on the framework's generalizability to other multimodal tasks beyond sentiment analysis.

**Questions:**

1. Can the authors provide more details on the computational requirements and training time of ReFA compared to baseline methods?
2. How does the performance of ReFA change with varying amounts of training data? Is there a minimum data requirement for the framework to be effective?
3. Have the authors explored the applicability of ReFA to other multimodal tasks beyond sentiment analysis? If not, what modifications might be needed?
4. Could the authors provide insights into why the language modality seems to be particularly effective in unimodal scenarios?
5. Are there any scenarios or types of data where ReFA might not perform well? It would be helpful to discuss potential limitations.
6. How does the framework handle noisy data within the available modalities? Can the authors provide experimental results or discussions on the impact of noisy data on the performance of ReFA?

**Limitations:**

The paper does not explicitly address limitations or potential negative societal impacts of the work. Some suggestions for improvement:
1. Societal Impact: While the paper mentions broader impacts and limitations in the appendix, the main text lacks a detailed discussion on potential negative societal impacts. The authors should consider elaborating on ethical concerns, such as the potential misuse of sentiment analysis in sensitive applications or privacy issues related to multimodal data collection.
2. Discuss potential biases in the datasets used and how they might affect the model's performance across different demographic groups.
3. Acknowledge any limitations in the generalizability of the results to real-world, non-curated data.

---

> ### Author Rebuttal · Authors · 2024-08-07
>
> **Q1**: About computational complexities.
>
> **A1**: Thank you for your comments! We need to clarify that we have compared and discussed the proposed framework with the existing methods in terms of the number of parameters, FLOPs, and performance in three testing situations, as shown in Section A.4 of the Appendix. The proposed framework has the strongest robustness against missing modalities but has the lowest FLOPs and a lower number of parameters than most of the methods, which achieves a reasonable trade-off between complexity and performance.
> ***
> **Q2**: About the technical contribution of the proposed components.
>
> **A2**: We need to clarify that:
> * Mutual information maximization and adversarial learning are not simply combined and employed but are specifically designed to address the problems and limitations of existing MSA methods under missing modality situations.  Specifically, existing methods lack effective supervision of semantic alignment and distributional alignment during feature reconstruction, resulting in the inability to precisely recover sentiment semantics, as stated in lines 37-38 & lines 87-90 & lines 157-160 & lines 184-188 in the main manuscript. In contrast, the proposed interaction paradigm guides the reconstruction of missing sentiment information both at the semantic level and at the distributional level, thus restoring realistic modality features as much as possible.
> * Our proposed components bring significant performance gains, which are essential in the framework. The ablation results in Section 4.4 in the main manuscript and Section A.3 in the Appendix demonstrate this advantage.
> ***
> **Q3**: Description of the models used for the classification or regression.
>
> **A3**: Thank you for your reminder! The models used for classification and regression are fully-connected layers, including two linear layers, a ReLU activation layer, and a Softmax layer (in the case of classification tasks). We promise to add detailed descriptions in the revision.
> ***
> **Q4**: Discuss the potential limitations or cases where it might not perform well.
>
> **A4**:
> *  We need to clarify that we have already discussed some of the limitations of this paper, i.e., the framework is based on teacher network trained on complete-modality data, and thus its performance depends to some extent on the upper bound of the teacher network's performance, as stated in A.6 in the Appendix.  In the future, we plan to use the self-distillation paradigm to improve the flexibility and applicability of the model.
> * Furthermore, in real-world applications, modality missing cases can be very intricate and complex, leading to a possible minor loss in model performance. In the future, we will explore more complex modality-missing cases to compensate for this deficiency. We will add this limitation in the revision.
> ***
> **Q5**: Discuss the framework generalization to other multimodal tasks.
>
> **A5**: Valuable suggestions! We added comparison experiments of our framework with some of the baselines on humor detection (i.e., UR-FUNNY dataset) and sarcasm discovery (MUSTARD dataset) tasks, as shown in Table. 1 in **reply.pdf**.
> The testing conditions include Intra-Modality Missingness (Intra-MM), Inter-Modality Missingness (Inter-MM), and Complete Modality (CM). The superior experimental results demonstrate the generalization of the framework to multiple multimodal tasks.
> ***
> **Q6**: About ReFA performance with data size.
>
> **A6:** The framework has a strong generalization to datasets of different sizes for the following reasons:
> * We have launched comprehensive experiments on the MOSI, IEMOCAP, and MOSEI datasets of sequentially increasing sizes, as shown in Section 4.3 in the main manuscript.
> * In addition, we have conducted experiments based on different ratios of samples in the MOSI dataset, as shown in Fig. 1 in **reply.pdf**. The experimental results are the average under five different random seeds.
> ***
> **Q7**: The reason for Language being effective in unimodal scenarios.
>
> **A7**: Constructive Comments. We provide two insights below:
> * Compared to linguistic modalities, non-linguistic modalities are potentially unreliable because audiovisual feature extraction tools typically introduce additional redundancy and noise.
> * As highly abstract symbolic systems, linguistic modalities typically contain more information and knowledge density, providing more effective structured semantic representations.
> ***
> **Q8**: About the noise effect on ReFA performance.
>
> **A8**:  We add various ratios of Gaussian noise to the MOSI dataset, and Fig. 2 in **reply.pdf** demonstrates the robustness of the proposed framework against noisy data, as the factorization mechanism adequately captures sentiment cues and filters out noise.
> ***
> **Q9**: About potential social impacts.
>
> **A9**: Thanks for the comments! We will add a detailed discussion of social implications to the main revision manuscript.
> ***
> **Q10**: About potential bias in the dataset.
>
> **A10**: In practice, we observe two possible biases in the datasets, including label bias and context bias.
> * The label bias usually occurs when the number of training samples for a specific category is more significant than for other categories. Such unbalanced data distribution would lead to trained models relying heavily on label bias as statistical shortcuts to make inaccurate predictions across demographic groups.
> * The context bias emerges when trained models exhibit strong spurious correlations between specific categories and context words in language modality. MSA models tend to predict samples containing those words to an incorrect category based on biased statistical information rather than intrinsic textual semantics.
> ***
> **Q11**: About result generalizability.
>
> **A11**: Non-curated data in real-world applications may contain more intricate cases of missing modalities, leading to a slight performance loss of the model. We will add this description in the revision.

---

> > ### Comment · Reviewer_zmpp · 2024-08-09
> > **Maintaining Positive Assessment After Thorough Rebuttal**
> >
> > Thank you for your comprehensive rebuttal. I appreciate the time and effort you have invested in addressing each point raised in my review. After carefully considering your responses, I still hold a positive assessment of the paper.
> >
> > Regarding computational complexity (Q1), I am pleased to see that you have included a comparison of parameters and FLOPs in the Appendix, which effectively addresses my concerns about computational requirements. On the topic of classification and regression models (Q3), I appreciate the details provided about the models used. Including this information in the main text will undoubtedly enhance clarity.
> >
> > I am also glad to hear that you will be adding discussions on social impacts and dataset bias. These additions will significantly improve the comprehensiveness of the paper. Your acknowledgment of potential performance loss in more complex real-world scenarios demonstrates a balanced perspective on the result's generalizability.
> >
> > Overall, your response has effectively addressed my concerns and questions. The additional experiments and planned revisions will further strengthen the paper, resulting in a more comprehensive and impactful contribution to the field of multimodality. In light of your thorough response and planned changes, I maintain my original assessment of the paper.

---

> > > ### Author Response · Authors · 2024-08-10
> > > **Response to Reviewer MG71 zmpp**
> > >
> > > Thank you for the valuable suggestions and recognition of our work.
> > >
> > > We promise to add the following to the revision: additional experiments, a description of the models used for classification or regression, a discussion of potential societal impacts, an analysis of potential biases in the dataset, and an acknowledgement of the generalizability of the results.
> > >
> > > We will endeavor to make more valuable contributions to the multimodal community.

---

### Author Rebuttal · Authors · 2024-08-07

We thank all reviewers for their suggestions and thoughtful comments.

---

### Decision · Program_Chairs · 2024-09-25

**Decision:**

Accept (poster)

**Comment:**

The paper studies the problem of incomplete multimodal sentiment analysis, where some modalities may be missing. The authors propose Representation Factorization and Alignment (ReFA) to tackle the issue of uncertain missing modalities in multimodal sentiment analysis (MSA). The proposed ReFA includes three components: Fine-grained Representation Factorization (FRF) module, Hierarchical Mutual Information (HMI) maximization mechanism, and Hierarchical Adversarial Learning (HAL) mechanism. Experiments are conducted on three datasets, demonstrating that the ReFA framework significantly improves MSA performance under both uncertain missing-modality and complete-modality testing conditions.

This paper received four reviews, including 2 strong accept, 1 accept, and 1 borderline accept. The reviewers recognized the novelty and contributions of this paper, such as practical issues, effective algorithms, comprehensive experiments, and good structure.

Meanwhile, the reviewers pointed out some concerns and suggestions to further improve the quality of this paper, including but not limited to: lack of a detailed discussion on the computational complexity and runtime performance of the proposed framework, missing discussion of potential limitations, incremental technical contributions, lack of differentiation in handling the unique characteristics of each type of information, lack of relevant case studies, and too brief problem definition. The presentation of this paper also needs to be improved.

The authors provided meaningful responses to the reviewers' comments and concerns. I recommend acceptance of this paper and strongly recommend the authors take the reviewers' comments and the promised revisions into consideration when preparing the final version.